# The influence of social capital in the utilisation of sexual reproductive health services among the youth in Ghana. A community-based cross-sectional study

**Mark Kwame Ananga**[1]*, **Rosemary Kafui Agbefu**[2], **Paul Narh Doku**[3], **Adom Manu**[4], **Deda Ogum Alangea**[4], **John Kumuori Ganle**[4]

1 Department of Population and Behavioural Sciences, Fred N. Binka School of Public Health, University of Health and Allied Sciences, Hohoe, Ghana, 2 Institute of Alternative and Traditional Medicine, University of Health and Allied Sciences, Ho, Ghana, 3 Department of Mental Health, School of Nursing and Midwifery, University of Cape Coast, Cape Coast, Ghana, 4 Department of Population, Family and Reproductive Health, School of Public Health, University of Ghana, Legon, Ghana

* kananga@uhas.edu.gh

**Data Availability Statement:** All relevant data are within the paper and its Supporting Information files.

## Abstract

Social capital, often seen as the resources accessed through social connections, is currently gaining much attention in public health. However, limited studies have focused on the relationship between social capital and reproductive health services. Besides, while the factors associated with the use of reproductive health services among the youth are well documented in the literature, most studies have focused on narrowed perspective failing to take cognisance of the role of social capital. Yet, it is known that these behaviours can be influenced by social factors, which may be beyond the individual's control partly because the youth are embedded in social organisations. Therefore, this study examined the relationship between social capital and the youth's utilisation of reproductive health services. The study population comprised adolescents and young adults aged 15–24 years who were both in and out of school at the time of the survey. The study used a cross-sectional quantitative design involving a community-based household survey method to sample 792 respondents through multi-stage cluster sampling. The chi-square test examined the relationship between sociodemographic, social capital variables and reproductive health services. To account for potential confounding factors, a multivariable logistic regression model included variables from the binary logistic regression analysis with a p-value less than 0.05. In general, access to higher social capital was observed among 493 (62.2%). Almost half, 385 (48.6%) of the respondents have ever used at least one of the reproductive services examined in this study. After controlling all other significant predictors, the following social capital variables remained associated with increased utilisation of SRH services: higher trust in neighbourhood (AOR = 1.8; CI = 1.22–2.66), higher trust in people/institutions (AOR = 2.66; CI = 1.82–3.99), higher social cohesion (AOR = 3.35; CI = 2.21–5.08), stronger network (AOR = 7.55; CI = 4.43–12.87). Access to some social capital dimensions is associated with increased use of reproductive health services. However, any intervention such as mentoring including peer support programs, meant to address social capital needs in sexual and

**Funding:** The authors received no specific funding for this work.

**Competing interests:** The authors have declared that no competing interests exist.

reproductive health should consider the efficacy of each social capital dimension and the intervention's environment.

## 1. Introduction

Young people in Sub-Saharan Africa confront numerous sexual and reproductive health challenges, minimal access to and utilisation of sexual and reproductive health services and education on sexuality and family planning (FP). This has led to high rates of sexually transmitted infections (STI) and HIV prevalence among young people, early pregnancy, and vulnerability to delivery problems, all of which have resulted in high rates of death and disability [1]. However, using sexual and reproductive health (SRH) services is not as prevalent as one might assume, given the efforts made in Africa. The utilisation of SRH services varies greatly, from as low as 7.9% for adolescent and youth-friendly services in Ghana [2] to 96.1% for SRH services in general in Ethiopia [3], indicating that different socio-environmental and contextual factors influence utilisation differently in other parts of the continent. Currently, most of the Ghana Health Service's public health facilities around the nation include Adolescent/Youth Health corners that provide sexual and reproductive services tailored to the unique requirements of adolescents/youth. In addition, several Non-Governmental Organisations are also active in delivering comprehensive adolescent health services, such as SRH information and the development of Adolescent/Youth Health Corners in health institutions and schools. Nonetheless, young people continue to underutilise sexual and reproductive healthcare resources [2]. Factors associated with utilisation of reproductive health services such as age, educational level, schooling status, good Knowledge of SRH services, discussion about SRH services, and educational status of partner have all been documented [4–6].

The concept of social capital has been used differently in economics, sociology and political science [7]. It has recently been an essential concept in public health [8]. Scholars have described social capital as social ties that may give people and communities access to resources and support in networks that are available in the local community [9]. The exchange of favours, the maintenance of group standards, trust towards individuals or organisations, and the provision of support to members of social groups are all examples of social reciprocity [9]. Many theories of social capital have their roots in individual and family property, but they are also growing to include ideas like communities and nations [10–12]. Both a structural and a cognitive form of explanation are viable options for social capital theory. In its structural form, it focuses on the aspects of social organisations that are visible from the outside and refers to the intensity of an individual's participation in community networks as measured in terms of objective criteria. This portion of the theory is observable from the outside [13]. Subjective factors such as norms and values are encompassed in the cognitive form; these subjective aspects can be measured subjectively [14, 15]. Not mutually exclusive, structural and cognitive types of social capital are characterised in terms of social interactions as what people "do" and what they "feel," respectively [14, 15].

Some scholars in the field of public health have focused their attention on the differentiation of social capital into "bonding," and "bridging," all of which are closely related to structural social capital [15, 16]. Establishing strong intra-group links that connect family members, neighbours, and close friends is an example of bonding capital. Bonding capital consists of relations inside groups of people who are similar to one another. The term "bridging capital"

refers to connections that are not very strong between persons or organisations with low levels of social participation.

Even though the evidence on social capital and health outcomes is growing, very little is known concerning the link between social capital and utilisation of sexual and reproductive health service in developing countries. Studies of how social capital influences knowledge and the use of sexual and reproductive healthcare services among the youth are scant. Few studies [17–19] conducted on social capital and utilisation of reproductive health services are limited to general populations and done outside sub-Saharan Africa. These studies indicate that bridging social capital was predominantly associated with promoting knowledge about available sexual and reproductive healthcare services [19]. In addition, having low trust in others has been identified as a precursor for poor knowledge and, ultimately, poor service utilisation. It has also been noted that lack of social support was associated with low HIV knowledge and not knowing where to seek HIV counselling and testing [18]. Having a cohesive family, a strong social network, and peers who encourage healthy behaviours and discussions of sexual heath help adolescents has also been noted to achieve a higher level of HIV knowledge among Vietnamese adolescents [17]. Available literature reveals that social capital is essential for improving health in settings with limited resources; nevertheless, additional study is required to discover the most appropriate ways to quantify social capital [20]. The current study took a comprehensive conceptualisation of social capital to understand the relationship between social capital and the youth's utilisation of sexual and reproductive health services.

## 2. Materials and methods

### 2.1. Study design and settings

The study employed a cross-sectional quantitative community-based household survey design. The study was conducted in the South Tongu District of the Volta region of Ghana. South Tongu district, with Sogakope as its capital, is located in the southern part of the Lower Volta Basin and bounded to the north by the Central and North Tongu Districts, to the east by the Akatsi South District, to the west by the Ada East District of the Greater Accra Region and to the south by the Keta Municipality. The District occupies a total land area of 643.57 square kilometres, representing 3.1% of the total land area of the Volta Region. The District has a population of 87,950 and this represent 4.1% of the total population of the Volta Region [21]. The district is mainly rural: 87.1% of the population resides in rural communities. The sex ratio, defined as the number of males to 100 females, is about 84 [21]. The population is generally very youthful, depicting a broad base pyramid with a small number of elderly people (60 years and above) 7.8%. According to the Ghana Statistical Service, with increasing age, the structure looks slightly thinner for the males than for the females, indicating that at older ages, the proportion of males is lower than that of females. At age 20–24 years, the proportion of males to females is the same. According to the District's Analytical Report, there is one Government District Hospital, one Catholic Mission hospital (Comboni Catholic Hospital), 13 Community Based Health Planning Zones, Four Health Centres, one Planned Parenthood Association of Ghana's (PPAG) Clinic and two private facilities [21].

### 2.2. Study population

The study population comprised adolescents and young adults aged 15–24 years who were both in and out of school. This age group was chosen because most youth within this group are currently sexually active [2].

## 2.3 Eligibility criteria

All persons between the ages of 15-24-year-olds were eligible. Participants must have lived in South Tongu District within the last twelve (12) months prior to the study. Before the research, participants must have been sexually active and should have given consent or assent to participate and acquired parental consent, if needed. All teens and young adults who met all inclusion criteria but had mental or psychological problems that prevented them from making an informed decision to participate in the study were excluded.

## 2.4. Sample size determination and sampling procedures

This study was part of a larger Doctoral Thesis on both risky sexual behaviours and utilisation of SRH services and thus sample size was derived from studies of two outcome variables: multiple sexual partners [22] and current contraceptive use [23]. Based on the calculations and assumptions below, the outcome variable (multiple sexual partners) that yielded the largest sample size estimated was used as the final sample size for the study. The assumptions and calculations is as follows:

95% confidence level with an alpha value of 0.05 ($\alpha$ = 0.05 value) yielding a Z $\alpha/2$ = 1.96 on a normal distribution curve. Prevalence of multiple sexual partners obtained in Ghana from a previous study was 33.5% [22]. Based on this assumption, the minimum sample size for the study was computed using the single population proportion formula as indicated below:

$$n = \frac{Z_{\frac{z}{2}}^2 p(1-p)}{e^2}$$

Where, n = Sample size.

z = the standard normal curve score value corresponding to the given confidence interval = 1.96.

$p$ = Prevalence of multiple sexual partners = 0.335.

$e$ = margin of error (the required precision) = 5%

$$n = \frac{Z_{\frac{z}{2}}^2 p(1-p)}{e^2} = (1.96)^2 * 0.335\,(1-0.335)/0.05^2 = \textbf{342.3}$$

A design effect size of 2 was taken into consideration in order to compensate the loss of sampling efficiency and enhance the variance of the parameter estimations, and therefore precision. In addition, a 15% non-response rate was assumed, taking into consideration the sensitive nature of the topic. Hence the final sample sizes were as follows:

342.3 * 2 +106.69 = 791.29 approximately **792**

Household-based survey was used as the main data collection method. The 2010 population and housing census document provided a list of twenty (20) communities that were included in the study. From these 20 communities, two (2) were classified as Urban whilst the rest were rural. The two urban communities were purposively selected and included in the study. A simple random sampling (by balloting) was used to select an additional eight rural communities from the rest to be included in the study. Based on the number of households per each community as provided by the 2010 population and housing census, sample size for each community was calculated proportional to the size of households in each community.

The next stage entailed selection of participants from households through a modified random walk [24]. The random walk method was the most appropriate because the research team could not get access to the household listing from the district statistical service and therefore random route walk was done using the housing units available.

Based on the modified random walk method, households were selected from housing units within the selected communities. In each community, a random starting point and the direction of travel was chosen. After determining a random starting point, Research Assistants entered the nearest house and all young people aged 15–24 years were identified. In each of the

houses, only one household was selected. For this purpose, all selected single-household houses with one eligible young person automatically qualified for interview. In houses with multiple households, however, a simple random balloting method was used to randomly select the household. Additionally, in households with more than one eligible person, one person was randomly selecting using the same simple balloting method.

## 2.5. Study variables

The outcome variable utilisation of reproductive health services was defined as using any of the minimum packages offered at the service delivery point, whether in a government or commercial facility, within the last year prior to the study. Measures of use included the use of the following: Family Planning and Counselling Services, Information and Contraceptives, testing and counselling for STIs and HIV, treatment and management of STIs and HIV, testing for pregnancy, women's pre- and post-abortion counselling.

The predictor variables for the study include the following social capital domains.

A number of independent variables were defined and measured, including different dimensions of social capital. In this study, social capital was defined in terms of groups and networks ties, trust that people have available to them for productive purposes, access to information and communication, social cohesion and empowerment. It was measured in terms of the following:

**i. Groups and networks.** There were eight (8) questions that measured groups and networks. Groups were assessed based on membership in any formal or informal group and associations to which research participants belonged. Other questions included the two most important groups, whether members shared same characteristics and whether these groups interacted with other groups (response consisted of 'YES', 'OCCASIONALLY' and 'NO'). To obtain strength of Bonding and Bridging Ties, the two most important groups and association were first defined for each participant. Bonding ties were derived from whether the group interacts with groups that share similar characteristics. "Bridging Ties" were derived from whether the groups interact with other groups that are different in terms of their characteristics. Strong 'bonding' or 'bridging' ties were obtained from scores on those who responded 'YES' to the above. Weak 'bonding' or bringing' ties were obtained from both OCCASIONALLY' and 'NO' responses.

Network size was measured by questions that bothered on the number of friends research participants had, and the number of people beyond his/her immediate family to call on in times of need. Responses were categorised into 0 = 'No one', 1 = 'One or two people', 2 = 'Three or four people' and 3 = 'Five or more people'. To obtain the network size row totals for number of friends a participant has, as well as, the number of people to call on when participants needed help, and those who can actually help the participants were calculated. Based on this, the maximum score was 6 and the minimum score from all these variables was 0. The median score was obtained and scores below the median score of 3 were categorised as small network size and score from 3 to 6 were classified as large network. The median was used to cater for any extreme value that may affect the normal distribution of the data.

**ii. Trust and Solidarity.** Five items made up the Trust and Solidarity scale. The first item generated information on generalised trust in the community: "Generally speaking, would you say that most people can be trusted"? The response was coded 0 = NO, and 1 = YES. The second question sought information on the extent to which participants agreed with the following statements on a Likert Scale of 1 to 5 where '1' = 'Disagree Strongly' and '5' = 'Agree Strongly'.

A. Most people who live in this village/neighbourhood can be trusted.

B. In this village/neighbourhood, one has to be alert or someone is likely to take advantage of you.

C. Most people in this village/neighbourhood are willing to help if you need it.

D. In this village/neighbourhood, people generally do not trust each other in matters of lending and borrowing money.

Reversed coding was done for questions B and D because in those questions, higher values rather denote lower trust.

To obtain "Trust in Neighbourhood", the aggregate score from A, B, C and D after reverse coding for questions B and D were used. The row total for A, B, C and D was calculated resulting in a minimum score of zero and maximum score of 4. Based on a median score of 2, "Higher Trust" was denoted by responses from 2 to 4 whilst lower trust was denoted by responses below 2. The final Trust in Neighbourhood was then dichotomised into High/Low Trust.

The next questions were used to elicit information on trust for institutions and officials in those institutions, such as government officials, doctors and nurses. For example, "On a scale of 1 to 5, where 1 means a very small extent and 5 means a very large extent, how much do you trust the people in that category?" People from your ethnic or linguistic group/race/caste/tribe, Local government officials, Central government officials, Teachers, and Nurses and doctors. Based on the scale above, a row total was obtained for trust in People from your ethnic or linguistic group/race/caste/tribe, Local government officials, Central government officials, Teachers, and Nurses and doctors. The minimum score was 5 should everyone disagreed to a very small extent and 25 if everyone agreed to a large extent. Based on this a median score of 14 was obtained. "Trust in People/Institutions" was obtained from dichotomising responses as "High" for scores 14 to 25 whilst scores below 14 were categorised into "Low Trust".

The score for "Solidarity" was obtained by asking participants, on a five-point scale, where 1 means always helping and 5 means never helping, "how well do people in your village/Neighbourhood help each other out these days? With a minimum score of 1 and a maximum score of 5, the median was used to categorise the responses as below 2 "Low Level" and 2–5 score being "High Level".

**iii. Information and Communication.** The score for Information and Communication was obtained from a 3-item question asking participants on a scale of 1 to 5, where 1 = never and 5 = everyday, how often they listen to radio, watch television, or visit any social media platform (Facebook, WhatsApp, twitter, Instagram). The minimum score for this variable was 3 and the maximum was 15. A median score of 8 was used to categorise the variable into "Low Usage (score 3–8) and "High Usage" (Score from 8 to 15).

**iv. Social Cohesion and Inclusion.** The score on "Social Cohesion" was measured by a four-item question. The first item sought information on whether the majority of people in this area generally have good relationships with each other. An initial 3 scale (1 = NO, 2 = Sometimes and 3 = YES) was later re-categorised into 0 (for 1) and 1(for 2 and 3) with zero indicating a lower score. The second item was also a five item Likert scale (1 = Very distant to 5 = very close) question, which measured how strong the feeling of togetherness or closeness in village/neighbourhood was. This was also recoded to 0 (for 1 to 2) and 1 (for 3 to 5). The third item sought information on whether participants felt that the area is theirs. An initial 3-point scale ranging from 1 = NO, 2 = Sometimes and 3 = YES was recoded into 0 (for 1) and 1 (for 2 and 3). The last item was a 5-point Likert scale indicating whether differences exist in the community in terms of religion or political views as well as sex differences, with 1 indicating to a very small extent and 5 = to a very greater extent. This was also recoded into 0 (for 3 to

5) and 1 (for 1 and 2). Based on the four items, a minimum score of zero indicating lower cohesion and 4 for higher cohesion was obtained. To categorise the variable into higher and lower cohesion, a median score was used, such that a score of 3 to 4 were categorised as "High" and score of 0 to 2 were categorised into "Low".

One item was used to obtain the score on "Social Inclusion". The question on social inclusion sought to find out if there were groups of people in the village/neighbourhood who were prevented from or do not have access to any of the following: Education/schools, Health services/clinics, Water, Justice, and Transportation. The response was categorical 1 = YES and 2 = NO. This was later recoded into 0 = Inclusive and 1 = Not Inclusive.

**v. Empowerment.** Three questions were used to measure empowerment. The first question bothered on how much control participants felt they had in making decisions that affect their everyday activities measured on a 5-point Likert scale (1 = No control– 5 = Control over all decisions). This was recoded into 0 (1–2) and 1 (3–5). The second question sought information on whether participants felt they had the power to make important decisions that change the course of their life (1 = Totally unable to change life– 5 = Totally able to change life). This was recoded into 0 (1–2) and 1 (3–5). The final question was on how much impact participants thought they had in making his/her village/neighbourhood a better place to live (1 = No impact, 2 = A small impact, 3 = A big impact). This was also recoded into 0 (1) and 1 (2 and 3). To compute the composite "Empowerment Score", the row total for the three recoded items were obtained together with a median. Scores above the median were categorised as "High Empowerment" while those below the median were categorised "Low Empowerment".

## 2.6. Data collection tool and procedure

An interviewer-administered questionnaire collected data on the Open Data Kit (ODK) platform containing sociodemographic data, social capital questions which were adapted from Social Capital Integrated Questionnaire (SC-IQ) for measuring social capital in low-income countries and questions on SRH services adapted from 2014 GDHS and 2017 GMHS. The data was collected by eight undergraduate students of public health (equal number of males and females who worked in pairs) and supervised by a public health nurse.

## 2.7. Data quality control

Training was provided to data collectors and supervisors about the purpose of the study, the questionnaire in detail, the procedure for collecting the data, the setting in which the data was collected, and the rights of study participants in order to guarantee the quality of the data. In order to conduct a pilot study of the questionnaire, thirty young people from Adidome, the capital of the Central Tongu District, were chosen at random. These thirty youths were not part of the actual research sample that was conducted. The modifications were made in response to the weakness that were discovered. During the entire process of data collection, supervisors and principal investigators checked the collected data for completeness, accuracy, and consistency. This was done both before and after each data collection session.

## 2.8. Data analysis and presentation

Data was extracted from the ODK and imported into Stata version 16 for analysis. The data was stratified based on sex and residence because these variables were found to have significant effect on both social capital and the use of reproductive health services under consideration. Frequencies and percentages were used to describe the distribution of research participants. At the bivariate level, chi square test was used to examine the relationship between the various sociodemographic as well as social capital variables and use of SRH services. Finally, in order

to account for the influence of any potential confounding factors, the variables from the binary logistic regression analysis that demonstrate an association and have a *p-value* that is either less than or equal to 0.05 were included in a multivariable logistic regression model. This implies that for each social capital variable under consideration, the researchers controlled for sociodemographic variables and other social capital variables that significantly influence the use of reproductive health services by young people. *P-value* less than or equal to 0.05 is considered as the level of significance.

## 2.9. Ethical approval

Ethical approval was obtained from the Ghana Health Service Ethics Review Committee **(GHS-ERC019/05/19)**. In addition, administrative permissions were also obtained from the Regional and District Directorates of Ghana Health Service in the Volta region and the South Tongu District before the commencement of the study.

**Consent.** For participants who were youngsters under 18, the participant's parent or legal guardian gave their written informed consent in addition to the adolescents' assent for their participation in the study. The participants in the study were explained the study's purpose, and data were collected only after obtaining their full informed written consent. Additionally, the participants' names and other personal identification information were removed to ensure the information remained confidential.

## 3. Results

A total 792 respondents were involved in the study yielding a 100% response rate

### 3.1. Sociodemographic characteristics of respondents

The mean age was 20.05 (± 2.49) years with age range between 15 and 24 years. Of the 972 respondents 58.59% were males, 67.05% came from rural area, 89.30% were never married. Almost two-thirds of the respondents 65.0% were out of school at the time of the study, 25.1% lived with both parents, 23.6% lived with mother alone or lived on their own 19.4%. Of all the respondents almost 60% possess a valid National Health Insurance card (*see* Table 1).

### 3.2. Access and distribution of social capital

Six hundred and thirty four (80.1%) respondents had access to large networks, 601 (75.9%) and 517 (65.3%) to strong bonding and bridging ties, respectively. 499 (63.0%) of respondents trusted their neighbourhood, while 466 (58.8%) trusted people and institutions, including health service providers. 478 (60.4%) respondents reported improved information and communication. 652 (82.3%) said their neighbourhood had low social inclusion. 560 (70.7%) respondents reported higher neighbourhood social cohesiveness. Finally, 477 (60.2%) respondents reported higher empowerment (see Table 2).

Results from Table 3 indicate that almost half (48.6%) of the respondents have ever used at least one of the reproductive services examined in this study. Majority (73.5%) reported using a condom.

When respondents were asked if they were aware of the existence of adolescent/youth corners in the district, only 144 (18.2%) responded in the affirmative. Out of the 144 respondents who were aware of the existence of adolescent/youth corners, 51.4% used the facility.

Significantly higher levels of awareness about AYFC were observed in urban areas than in rural areas $\{X^2 (1) = 11.82, p<0.001\}$. Out of the 144 respondents who were aware of the existence of adolescent/youth corners, 51.4% (74) actually made use of the facility. Higher

**Table 1. Demographic characteristics of respondents.**

| Variable | Frequency | Percentage |
|---|---|---|
| **Age** | | |
| **15-19yrs** | 343 | 43.3 |
| **20-24yrs** | 449 | 56.7 |
| *Mean age (SD) 20.04 (2.49)* | | |
| **Sex** | | |
| **Male** | 464 | 58.6 |
| **Female** | 328 | 41.4 |
| **Residence** | | |
| **Urban** | 261 | 32.9 |
| **Rural** | 531 | 67.1 |
| **Schooling Status** | | |
| **Out of School** | 515 | 65.0 |
| **In School** | 277 | 35.0 |
| **Marital Status** | | |
| **Single/never married** | 707 | 89.3 |
| **Married** | 36 | 4.5 |
| **Co-habiting** | 49 | 6.2 |
| **Ethnic Group** | | |
| **Ewe** | 759 | 95.8 |
| **Ga/Dangme** | 15 | 1.9 |
| **Akan** | 13 | 1.6 |
| **Others** | 5 | 0.6 |
| **Religion** | | |
| **None** | 12 | 1.5 |
| **Christianity** | 753 | 95.1 |
| **Muslim** | 18 | 2.3 |
| **Traditionalist** | 6 | 0.8 |
| **Others** | 3 | 0.4 |
| **Living with** | | |
| **Both Parents** | 199 | 25.1 |
| **Mother Alone** | 187 | 23.6 |
| **Father Alone** | 42 | 5.3 |
| **Grandparents** | 65 | 8.2 |
| **Siblings** | 34 | 4.3 |
| **Step Parents** | 4 | 0.5 |
| **Spouse /Partner** | 87 | 11.0 |
| **Guardian (Non-Family)** | 20 | 2.5 |
| **On their Own** | 154 | 19.4 |
| **Occupation** | | |
| **Government Worker** | 13 | 1.6 |
| **Traders** | 62 | 7.8 |
| **Artisans** | 175 | 22.1 |
| **Farmer/Fisherman** | 17 | 2.2 |
| **Drivers/Riders** | 94 | 11.9 |
| **Students** | 277 | 35.0 |
| **Unemployed** | 123 | 15.5 |
| **Others** | 31 | 3.9 |

*(Continued)*

**Table 1.** (Continued)

| Variable | Frequency | Percentage |
|---|---|---|
| **Father's Education** | | |
| No Formal Education | 100 | 12.6 |
| Primary | 62 | 7.8 |
| JSS/JHS/Middle School | 241 | 30.4 |
| SSS/SHS | 123 | 15.5 |
| Tertiary | 40 | 5.1 |
| Don't Know | 226 | 28.5 |
| **Mother's Education** | | |
| No Formal Education | 146 | 18.4 |
| Primary | 121 | 15.3 |
| JSS/JHS/Middle School | 224 | 28.3 |
| SSS/SHS | 73 | 9.2 |
| Tertiary | 17 | 2.2 |
| Don't Know | 211 | 26.6 |
| **Had Valid NHI Card** | | |
| Yes | 472 | 59.6 |
| No | 320 | 40.4 |

utilisation was observed among urban dwellers in terms of use of adolescent/youth corners {$X^2$ (1) = 6.48, $p<0.011$}.

Reasons for non-utilisation of adolescent/youth corners ranged from fear of parents to feeling too young for the services perceived lack of privacy at the adolescent corners, not knowing the actual location of service providers, and perceived long waiting time (see Fig 1).

Most social capital variables were significantly associated with the use of sexual and reproductive health services at the bivariate level (See Table 4). After controlling for area of residence, schooling status, father's education, marital status, age, distance to health facility and valid health insurance, the following social capital variables remained statistically significant. High empowerment was associated with 35% reduced odds of using SRH services (AOR = 0.65; p = 0.026) compared to low empowerment. Stronger bridging ties were associated with about 58% lower odds of using SRH services (AOR = 0.42; p = 0.003). However, a stronger network was associated with almost 8 times higher likelihood of using SRH services (AOR = 7.55; p<0.001). Respondents with higher neighbourhood trust were 1.8 times more likely to use SRH services (AOR = 1.80; p <0.001). Similarly, participants who reported higher trust in people/institutions were 2.66 times more likely to use SRH services (AOR = 2.66; p <0.001) than those who reported lower trust. Participants who reported living in areas with higher social cohesion were 3.35 times more likely to use SRH services (AOR = 3.35; p<0.001). Social cohesion has the largest effect size (0.49 or 49%) among all covariates that were significant predictors of sexual and reproductive health service utilisation. All the effects size were statistically significant.

Table 5 shows bivariate and multivariate analyses of social capital and Adolescent and Youth Friendly Corner use (AYFC). Three social capital variables, bridging ties, network of relationship and access to information and communication, predicted AYFC use at the bivariate level. After controlling for area of residence, schooling status and distance to health facility, bridging ties no longer predicted AYFC use. Participants with stronger networks were four times more likely to use AYFC (AOR = 4.08; p = 0.012) than weaker networks. Access to High information and communication was associated with almost two times the odds of AYFC use (AOR = 1.97; p = 0.036) compared to low access.

**Table 2. Access and distribution of social capital.**

| Variable | Total (%) | Sex | | | Residence | | |
|---|---|---|---|---|---|---|---|
| | | Male (%) | Female (%) | $x^2$ (p-value) | Urban (%) | Rural (%) | $x^2$ (p-value) |
| **Network size** | | | | | | | |
| Small | 158 (21.0) | 98 (21.1) | 60 (18.3) | 0.96 (0.327) | 51 (19.5) | 107 (20.2) | 0.04 (0.840) |
| Large | 634 (80.1) | 366 (78.9) | 268 (81.7) | | 210 (80.5) | 424 (79.9) | |
| **Bridging Ties** | | | | | | | |
| Weak | 275 (34.7) | 159 (34.3) | 116 (35.4) | 0.10 (0.749) | 123 (47.1) | 152 (28.6) | **26.43 (<0.001)** |
| Strong | 517 (65.3) | 305 (65.7) | 212 (64.6) | | 138 (52.9) | 379 (71.4) | |
| **Bonding Ties** | | | | | | | |
| Weak | 191 (24.1) | 100 (21.6) | 91 (27.7) | **4.03 (0.045)** | 87 (33.3) | 104 (19.6) | 18.07 (<0.001) |
| Strong | 601 (75.9) | 364 (78.5) | 237 (72.3) | | 174 (66.7) | 427 (80.4) | |
| **Trust in Neighbourhood** | | | | | | | |
| Low Trust | 293 (37.0) | 186 (40.0) | 107 (32.6) | **4.59 (0.032)** | 90 (34.5) | 203 (38.2) | 1.05 (0.305) |
| High Trust | 499 (63.0) | 278 (59.9) | 221 (67.4) | | 171 (65.5) | 328 (61.8) | |
| **Trust in people/institutions** | | | | | | | |
| Low Trust | 326 (41.2) | 201 (43.3) | 125 (38.1) | 2.15 (0.142) | 103 (39.5) | 223 (42.0) | 0.46 (0.496) |
| High Trust | 466 (58.8) | 263 (56.7) | 203 (61.9) | | 158 (60.5) | 308 (58.0) | |
| **Level of solidarity** | | | | | | | |
| Low | 165 (20.8) | 101 (21.8) | 64 (19.5) | 0.59 (0.441) | 66 (25.3) | 99 (18.7) | **4.68 (0.030)** |
| High | 627 (79.2) | 363 (78.2) | 264 (80.5) | | 195 (74.7) | 432 (81.4) | |
| **Social Cohesion** | | | | | | | |
| Low | 232 (29.3) | 135 (29.1) | 97 (29.6) | 0.02 (0.884) | 63 (24.1) | 169 (31.8) | 4.99 (**0.025**) |
| High | 560 (70.7) | 329 (70.9) | 231 (70.4) | | 198 (75.9) | 362 (68.2) | |
| **Social Inclusion** | | | | | | | |
| Low | 652 (82.3) | 395 (85.1) | 257 (78.4) | **6.06 (0.014)** | 209 (80.1) | 443 (83.4) | 1.35 (0.245) |
| High | 140 (17.7) | 69 (14.9) | 71 (21.7) | | 52 (19.9) | 88 (16.6) | |
| **Information and Communication** | | | | | | | |
| Low | 314 (39.7) | 168 (36.2) | 146 (44.5) | **5.54 (0.019)** | 81 (31.0) | 233 (43.9) | 12.07 (< **0.001**) |
| High | 478 (60.4) | 296 (63.8) | 182 (55.5) | | 180 (69.0) | 298 (56.1) | |
| **Empowerment** | | | | | | | |
| Low | 315 (39.8) | 182 (39.2) | 133 (40.5) | 0.14 (0.708) | 133 (51.0) | 182 (34.3) | 20.33 (<**0.001**) |
| High | 477 (60.2) | 282 (60.8) | 195 (59.5) | | 128 (49.0) | 349 (65.7) | |

## 4. Discussion

In the current study, less than half of the respondents had used any reproductive health services offered to young people. The rate of utilisation of sexual and reproductive health services found in the current study is lower than the almost 70% among young people in Ghana [25] and 54% utilisation rate in Kumbungu District in Northern Ghana [26]. Other findings from Oromia, Amhara and the Northeast regions of Ethiopia reported higher utilisation of SRH services among young people than in the current study [3, 27, 28]. Differences in the populations involved could explain the lower utilisation rate found in the present study. Whilst most previous studies were mainly school-based, the current research involved in-school and out-of-school young people. Whilst in-school adolescents are likely to benefit from the various life skills training available through the school curriculum, that may not be available to out-of-school research participants.

Although the participants mentioned condoms, the rate of family planning and VCT utilisation were alarmingly low. The use of family planning is essential to avoiding unintentional

**Table 3. Utilisation of SRH services and adolescent/youth friendly corners.**

| Variable | Total (%) | Sex | | | Residence | | |
|---|---|---|---|---|---|---|---|
| | | Male (%) | Female (%) | $x^2$ (p-value) | Urban (%) | Rural (%) | $x^2$ (p-value) |
| **Use of SRH Services (n = 792)** | | | | | | | |
| Yes | 385 (48.6) | 231 (49.8) | 154 (47.0) | 0.62 (0.432) | 151 (57.9) | 234 (44.1) | 13.31 (**p<0.001**) |
| No | 407 (51.4) | 233 (50.2) | 174 (53.1) | | 110 (42.2) | 297 (55.9) | |
| **SRH Services Utilised (n = 385)** | | | | | | | |
| Condom Provision | 283(73.5) | 172(74.5) | 111 (72.1) | 7.57 (0.271) ⋔ | 112(74.2) | 283(73.5) | 4.97 (0.547) ⋔ |
| Family Planning | 47 (12.2) | 25(10.8) | 22 (14.3) | | 15(9.9) | 47 (12.2) | |
| VCT Services | 22 (5.7) | 14(6.1) | 8(5.19) | | 8(5.3) | 22(5.7) | |
| Pregnancy Testing | 15 (3.9) | 9 (3.9) | 6(3.9) | | 7(4.6) | 15 (3.9) | |
| ANC | 11 (2.9) | 5(2.16) | 6 (3.9) | | 7(4.6) | 11(2.7) | |
| Treatment of STIs | 6 (1.6) | 6(2.6) | 0 (0) | | 2(1.32) | 6(1.6) | |
| Pap Smear Screening | 1 (0.3) | 0 (0) | 1 (0.65) | | 0 (0) | 1(0.3) | |
| **Awareness about Adolescent/Youth Corners (n = 792)** | | | | | | | |
| No | 648 (81.8) | 380 (81.9) | 268 (81.7) | 0.01 (0.946) | 196 (75.1) | 452 (85.1) | 11.82 (**<0.001**) |
| Yes | 144 (18.2) | 84 (18.1) | 60 (18.3) | | 65 (24.9) | 79 (14.9) | |
| **Source of Information (n = 144)** | | | | | | | |
| Media (Mass & Social) | 69 (47.9) | 39 (46.4) | 30 (50) | 4.14 (0.68) ⋔ | 25 (38.5) | 44 (55.7) | 14.51 (**0.024**) ⋔ |
| School | 32 (22.2) | 19 (22.6) | 13 (21.7) | | 17 (26.2) | 15 (19.0) | |
| Health worker | 26 (18.1) | 15 (17.9) | 11(18.3) | | 16 (24.6) | 10 (12.7) | |
| Friends | 9 (6.3) | 4 (4.8) | 5 (8.3) | | 3 (4.6) | 6 (7.6) | |
| Family | 4(2.8) | 3(3.6) | 1 (1.7) | | 4 (6.2) | 0 (0) | |
| **Use of Youth Corners (n = 144)** | | | | | | | |
| No | 70 (48.6) | 37 (44.1) | 33 (55) | 1.68 (0.195) | 24 (36.92) | 46 (58.23) | 6.48 (**0.011**) |
| Yes | 74 (51.4) | 47 (56.0) | 27 (45) | | 41 (63.08) | 33 (41.77) | |
| **Frequency of Use (n = 74) Past 12 months** | | | | | | | |
| Only Once | 20 (27.0) | 15 (31.9) | 5 (18.5) | 6.43 (0.092) | 11 (26.8) | 9 (27.3) | 7.30 (0.063) ⋔ |
| Two or Three Times | 25 (33.8) | 13 (27.7) | 12 (44.4) | | 9 (22.0) | 16 (48.5) | |
| Four or Five Times | 13 (17.6) | 6 (12.8) | 7 (25.9) | | 9 (22.0) | 4 (12.1) | |
| Six and above Times | 16 (21.6) | 13 (27.7) | 3 (11.1) | | 12 (29.3) | 4 (12.1) | |

⋔ p-*value from fisher's exact*

pregnancy and hence decreasing unsafe abortion. The low utilisation of family planning in the current research could thus lead to increase rates of unsafe abortions because most participants in the present study were unmarried, albeit sexually active. Teenagers and young people, especially those in school, are at greater risk of the effect that inadequate sexual and reproductive health services can have on their reproductive health, including STIs, unwanted pregnancy and unsafe abortion.

While inadequate use of services may be a sign of young people's poor knowledge about sexual and reproductive health services, the fact that utilisation of AYFCs was very low calls into question the effectiveness of the adolescent corners concept and adolescent health policy more generally. In this study, many young people referred to the fear of parents, age inappropriateness, and lack of privacy at the youth-friendly facilities as their reasons for not using these services. Although these reasons are consistent with findings from a school-based cross-sectional study among Ethiopian youth [3], the results are concerning. For instance, fear of parental reproach concerning the use of youth corners noted in the current study is a cause for concern. Parents should therefore be made aware of the SRH needs of their adolescents and the services available to support them.

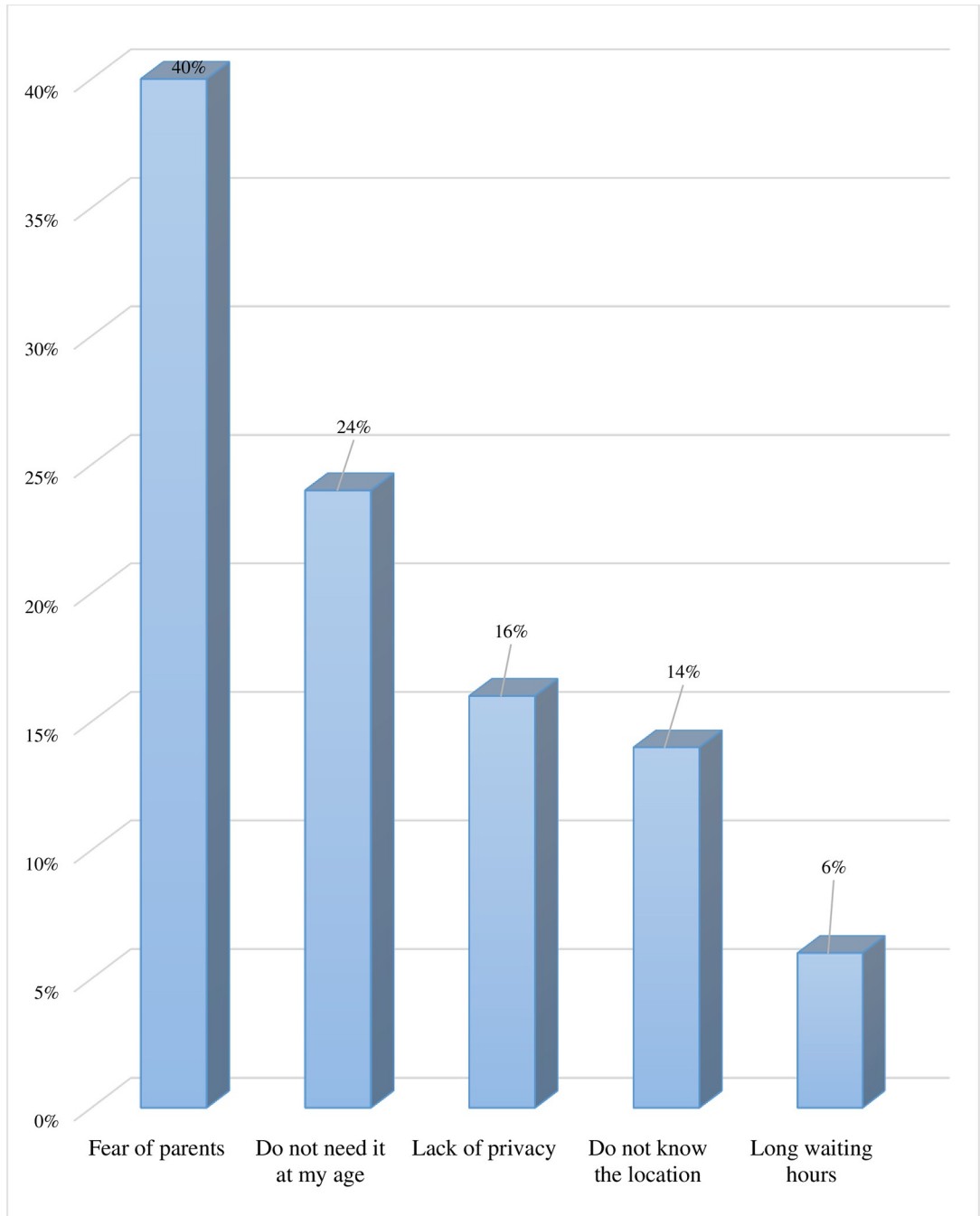

**Fig 1. Reasons for non-utilisation of AYFC among the youth.**

Findings from the current study indicate that the use of SRH services is positively associated with the following social capital variables in adjusted models: a stronger social network of connections in the community, trust in the neighbourhood, trust in people and institutions, higher bonding relationships and higher levels of social cohesion in the neighbourhood. In addition, the influence of social networks on reproductive health services was consistent with findings from earlier studies [17, 18, 29]. The finding from the current study is not surprising since social networks enable sharing of information on sexual and reproductive health services.

**Table 4. Association between social capital variables and utilisation of SRH services.**

| Social Capital Variable | Utilisation of SRH services | | | | | | | |
|---|---|---|---|---|---|---|---|---|
| | No (407) | Yes (385) | Crude Odds Ratio | | Adjusted Odds Ratio | | Effect size | |
| | n (%) | n (%) | OR (95% CI) | P-value | AOR (95% CI) | P-value | Effect size | P-value |
| **Network of Connection [a]** | | | | | | | | |
| Weak | 131 (32.19) | 27 (7.01) | Ref | | | | | |
| Strong | 276 (67.81) | 358 (92.99) | 6.29 (4.04 9.80) | <0.001 | 7.55 (4.43 12.87) | <0.001 | 0.25 | <0.00001 |
| **Trust Neighbourhood [b]** | | | | | | | | |
| Low | 199 (48.89) | 94 (24.42) | Ref | | | | | |
| High | 208 (51.11) | 291 (75.58) | 2.96 (2.19 4.01) | <0.001 | 1.8 (1.22 2.66) | 0.003 | | |
| **Trust People/Institution [c]** | | | | | | | | |
| Low | 208 (51.11) | 118 (30.65) | Ref | | | | | |
| High | 199 (48.89) | 267 (69.35) | 2.37 (1.77 3.16) | <0.001 | 2.66 (1.82 3.89) | <0.001 | 0.2447 | <0.00001 |
| **Information and Communication** | | | | | | | | |
| Low | 188 (46.19) | 126 (32.73) | | | | | | |
| High | 219 (53.81) | 259 (67.27) | 1.76 (1.32 2.35) | <0.001 | 1.01 (0.69 1.45) | 0.973 | 0.2046 | <0.00001 |
| **Empowerment [d]** | | | | | | | | |
| Low | 126 (30.96) | 189 (49.09) | Ref | | | | | |
| High | 281 (69.04) | 196 (50.91) | 0.47 (0.35 0.62) | <0.001 | 0.65 (0.45 0.95) | 0.026 | 0.1813 | 0.0014 |
| **Bridge Ties [e]** | | | | | | | | |
| Weak | 100 (24.57) | 175 (45.45) | Ref | | | | | |
| Strong | 307 (75.43) | 210 (54.55) | 0.39 (0.29 0.53) | (0.001 | 0.42 (0.24 0.75) | 0.001 | 0.2088 | 0.0006 |
| **Bond Ties** | | | | | | | | |
| Weak | 97 (23.83) | 94 (24.42) | | | | | | |
| Strong | 310 (76.17) | 291 (75.58) | 0.96 (0.69 1.34) | 0.848 | | | | |
| **Level of solidarity** | | | | | | | | |
| Low | 90 (22.11) | 75 (19.48) | | | | | | |
| High | 317 (77.89) | 310 (80.52) | 1.17 (0.83 1.66) | 0.362 | | | | |
| **Social Inclusion** | | | | | | | | |
| Low | 343 (84.28) | 309 (80.26) | | | | | | |
| High | 64 (15.72) | 76 (19.74) | 1.32 (0.91 1.90) | 0.139 | | | | |
| **Social cohesion [f]** | | | | | | | | |
| Low | 173 (74.57) | 59 (25.43) | Ref | | | | | |
| High | 234 (41.99) | 326 (58.21) | 4.09 (2.91 5.74) | <0.001 | 3.35 (2.21 5.08) | <0.001 | **0.4914** | **<0.00001** |

[a] adjusted for area of residence, schooling status, father's education, marital status, age, distance to facility, valid health insurance, trust in neighbourhood, trust in institution/people, information and communication, bridging ties, and social cohesion.

[b] adjusted for area of residence, schooling status, father's education, marital status, age, distance to facility, valid health insurance, network of connection, trust in institution/people, information and communication, bridging ties, empowerment, and social cohesion.

[c] adjusted for area of residence, schooling status, father's education, marital status, age, distance to facility, valid health insurance, network of connection, trust in neighbourhood, information and communication, empowerment, bridging ties,and social cohesion.

[d] adjusted for area of residence, schooling status, father's education, marital status, age, distance to facility, valid health insurance, network of connection, trust in institution/people trust in neighbourhood, information and communication, bridging ties, and social cohesion.

[e] adjusted for area of residence, schooling status, father's education, marital status, age, distance to facility, valid health insurance, network of connection, trust in neighbourhood, trust in institution/people, information and communication, empowerment, and social cohesion.

[f] adjusted for area of residence, schooling status, father's education, marital status, age, distance to facility, valid health insurance, trust in neighbourhood, trust in institution/people, information and communication, empowerment, bridging ties,and network of connection

**Table 5. Association between social capital and utilisation of adolescent/youth corners).**

| Social Capital Variable | Use of Adolescent/Youth Corners | | | | | | | |
|---|---|---|---|---|---|---|---|---|
| | No (70) | Yes (74) | Unadjusted | | Adjusted | | Effect size | |
| | n (%) | n (%) | OR (95% CI) | P-value | AOR (95% CI) | P-value | | P-value |
| **Network of Connection** [a] | | | | | | | | |
| Weak | 19 (27.14) | 9 (12.16) | Ref | | | | | |
| Strong | 51 (72.86) | 65 (87.84) | 2.29 (1.12 6.45) | **0.026** | 4.08 (1.36 12.28) | **0.012** | 0.1498 | **0.374** |
| **Trust Neighbourhood** | | | | | | | | |
| Low | 38 (54.29) | 37 (50.00) | Ref | | | | | |
| High | 32 (45.71) | 37 (50.00) | 1.18 (0.61 2.28) | 0.607 | | | | |
| **New Trust People/Institution** | | | | | | | | |
| Low | 39 (55.71) | 36 (48.65) | Ref | | | | | |
| High | 31 (44.29) | 38 (51.35) | 1.32 (0.68 2.55) | 0.397 | | | | |
| **Information and Communication** [b] | | | | | | | | |
| Low | 29 (41.43) | 19 (25.68) | Ref | | | | | |
| High | 41 (58.57) | 55 (74.32) | 2.05 (1.01 4.15) | **0.045** | 1.97 (1.04 4.78) | **0.036** | 0.1575 | **0.264** |
| **Empowerment** | | | | | | | | |
| Low | 23 (32.86) | 26 (35.14) | Ref | | | | | |
| High | 47 (67.14) | 48 (64.86) | 0.9 (0.45 1.80) | 0.773 | | | | |
| **Bridge Ties** [c] | | | | | | | | |
| Weak | 33 (47.14) | 23 (31.08) | Ref | | | | | |
| Strong | 37 (52.86) | 51 (68.92) | 1.98 (1.00 3.90) | **0.048** | 2.26 (0.92 5.43) | 0.068 | | |
| **Bond Ties** | | | | | | | | |
| Weak | 28 (40.00) | 22 (29.73) | Ref | | | | | |
| Strong | 42 (60.00) | 52 (70.27) | 1.57 (0.78 3.14) | 0.197 | | | | |
| **Level of solidarity** | | | | | | | | |
| Low | 21 (30.00) | 18 (24.32) | Ref | | | | | |
| High | 49 (70.00) | 56 (75.68) | 1.33 (0.63 2.78) | 0.444 | | | | |
| **Social Inclusion** | | | | | | | | |
| Low | 70 (100.00) | 74 (100.00) | ****** | | | | | |
| High | | | | | | | | |

****** All respondents in the category reported low social inclusion

[a] adjusted for area of residence, schooling status, distance to facility, information and communication, and bridging ties

[b] adjusted for area of residence, schooling status, distance to facility, network of connection, bridging ties

[c] adjusted for area of residence, schooling status, distance to facility, information and communication and network of connection

Such information becomes vital in young people's decisions concerning the use of sexual and reproductive services. For young people in rural communities, social networks are considered the primary source of information as they are severely disadvantaged in several aspects of their lives [30]. Higher trust in people/institutions and neighbourhoods was also associated with utilising reproductive health services in the current study. This finding is consistent with results from previous studies [18, 29, 31]. The findings also support the argument that trust in healthcare providers is critical in explaining healthcare usage [32].

These findings have policy implications for promoting healthy sexual and reproductive health development in more rural communities. Promoting diverse and more robust networks, including those that involve decision-makers, can provide adolescents better access and more opportunity to articulate their reproductive and sexual health needs and negotiate support for such services. Establishing and extending these various networks will benefit, most

importantly, underprivileged young people and adolescents with little assets and limited access to these services, thereby promoting equitable access to sexual and reproductive health services.

The findings suggest a need for health policymakers, community health organisations, and health administrators to consider the issue of trust when designing reproductive health-related interventions for vulnerable populations such as adolescents and young adults.

In the current study, higher bonding ties were associated with the use of sexual and reproductive health services. This finding is not consistent with previous studies [20, 33]. Research has shown that communities with higher bonding ties can have negative health consequences, particularly for poorer communities [33]. Therefore if the existing norms and values discourage reproductive health services, more impoverished communities with higher bonding ties may have lower utilisation of services. The difference in findings between the current and previous studies could be attributed to the population of interest and the norms and values shared within these groups. Therefore, it will be safe to say that the finding in the current study may be related to the spread of positive norms and values among the younger generation concerning the use of reproductive health services within their groups.

This suggests that involvement in groups, such as religious, social or school clubs, could encourage the youth to use SRH services. The findings would mean that building on social clubs associated with reproductive health could be critical in improving adolescents' and young adults' reproductive health beliefs and health information to encourage SRH service utilisation. There is also a need to collaborate with these social clubs since they benefit health beliefs and health information, which are essential in utilisation of health services.

### 4.1. Strength and limitations

Given that the concept of social capital is relatively new in public health, this study contributes to the literature on social capital and how it relates to young people's sexual and reproductive health. This research utilised modified versions of conventional questionnaires on social capital and reproductive health services, contributing to the study's high validity. The study employed a sufficient sample size and achieved a reasonable response rate. The investigators put in a lot of work to ensure that the data remained accurate, primarily by administering a pre-test, conducting numerous field supervisions, and providing training to the data collectors. Nevertheless, there are likely some limitations in the study. Social desirability and recollection bias may be introduced during the data collection process. In addition, residual confounders are noted here. Parental income and parental social capital in itself were not accounted, and this may have an influence on SRH utilisation by young people. Further, we noted as a limitation of our study, the fact that the questionnaire was not developed and validated specifically for our setting, it may not comprehensive enough to consider the context of Ghana. Finally, since this study was cross-sectional, it did not demonstrate any cause-effect linkages.

### 5. Conclusion

Whilst social capital could provide an essential resource for meeting adolescents sexual and reproductive health needs in Ghana, an understanding of which form of social capital has potential benefit for a particular aspect of sexual and reproductive health is very important.

### Supporting information

**S1 Questionnaire. Contains question that were used for the study.**
(PDF)

**S1 Dataset.**
(XLS)

## Author Contributions

**Conceptualization:** Mark Kwame Ananga, John Kumuori Ganle.

**Data curation:** Mark Kwame Ananga.

**Formal analysis:** Mark Kwame Ananga.

**Investigation:** Mark Kwame Ananga.

**Methodology:** Mark Kwame Ananga, John Kumuori Ganle.

**Project administration:** Rosemary Kafui Agbefu.

**Supervision:** Adom Manu, Deda Ogum Alangea, John Kumuori Ganle.

**Writing – original draft:** Mark Kwame Ananga, Rosemary Kafui Agbefu, Paul Narh Doku, Adom Manu, Deda Ogum Alangea, John Kumuori Ganle.

**Writing – review & editing:** Mark Kwame Ananga, Rosemary Kafui Agbefu, Paul Narh Doku, Adom Manu, Deda Ogum Alangea, John Kumuori Ganle.

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
