## [Decision Letter · Decision Letter 0]

13 Dec 2022

PGPH-D-22-01611

The influence of social capital in the utilisation of sexual reproductive health services among the youth in Ghana. A community-based cross-sectional study.

Dear Dr. Ananga,

Thank you for submitting your manuscript to PLOS Global Public Health. After careful consideration, we feel that it has merit but does not fully meet PLOS Global Public Health’s publication criteria as it currently stands. Therefore, we invite you to submit a revised version of the manuscript that addresses the points raised during the review process.

We look forward to receiving your revised manuscript.

Kind regards,

Orvalho Augusto, MD, MPH

Academic Editor

Journal Requirements:

1. Please provide separate figure files in .tif or .eps format only and remove any figures embedded in your manuscript file. Please also ensure that all files are under our size limit of 10MB.

2. Please amend your Data Availability Statement and indicate where the data may be found

Additional Editor Comments (if provided):

Reviewers' comments:

Reviewer's Responses to Questions

**Comments to the Author**

1. Does this manuscript meet PLOS Global Public Health’s publication criteria? Is the manuscript technically sound, and do the data support the conclusions? The manuscript must describe methodologically and ethically rigorous research with conclusions that are appropriately drawn based on the data presented.

Reviewer #1: Partly

Reviewer #2: Partly

2. Has the statistical analysis been performed appropriately and rigorously?

Reviewer #1: I don't know

Reviewer #2: Yes

3. Have the authors made all data underlying the findings in their manuscript fully available (please refer to the Data Availability Statement at the start of the manuscript PDF file)?

Reviewer #1: No

Reviewer #2: Yes

4. Is the manuscript presented in an intelligible fashion and written in standard English?

Reviewer #1: Yes

Reviewer #2: Yes

5. Review Comments to the Author

Reviewer #1: This is a cross sectional study that reports associations between various indicators of social capital and use of reproductive health services in a sample of 792 individuals aged 15-24 years in a district of the Volta region in Ghana.

This is cross-sectional study and causation cannot be determined. Therefore, the paper should avoid causal language that appears in several places, such as in Abstract “led to increased use…”.

It is not possible to assess the quality and validity of the indicators because the questions used to create the different predictor variables are not available. The reader would need to know at least 1) what questions were used for which indicators, 2) exactly how the variables were combined to create a score and 3) the cut-offs used to categorize into low and high levels for each social capital variable. In addition, the questionnaire should be available, at least as a supplementary file.

Results: The text repeats what’s shown in the tables. Please avoid redundancy by trimming the text.

Tables 2 and 3. Please provide a rationale for the stratification by sex and rural residence in the methods.

Figure 1 is not needed since this information can easily be included in Table 3

It is not clear how the modelling strategy and choice of covariates were determined. Abstract refers to adjustment for sex, marital status, age and access to valid National Health Insurance Card but the footnotes of Table 4 suggest more variables were included. The data analysis section in the methods is very general and does not provide enough details.

Residual confounding should be noted in the limitations.

Reviewer #2: This study investigates an interesting topic: the role of social capital in accessing youth sexual and reproductive health services in one district where such access remains particularly low in Ghana. The authors conducted a quasi-random population-level survey to assess the factors of low use of SRH services, an explored the role of social capital in facilitating access.

However, several weaknesses remain. First the article analyses social capital in the frame of the diverse obstacles which persist in accessing youth friendly SRH services. Improving on young people's social capital is presented as an alternative policy option, which can make up for service failure and stigma in the community. This proposition was trendy for a while at the beginning of the years 2000, in particular in World Bank publications, but never gained momentum. Scarce funds for youth SRH services are indeed probably better invested elsewhere than in building individuals' social capital.

I thus strongly suggest to focus the article *only* on social capital as a factor of access to youth SRH services. To do so, a better review of the literature on social capital and SRH is needed. As opposed to what is written, the literature on social capital in public health is huge. See for example:

Ehsan, Annahita, et al. "Social capital and health: A systematic review of systematic reviews." SSM-population health 8 (2019): 100425.

The article should summarize this vast literature and its key concepts. Some of the items like "empowerment" seem fully out of the scope of what is usually meant by social capital. A conceptual framework should then be proposed of the links of the different facets of social capital and SRH service utilization in particular. See for example:

Ireland, Hannah, Nguyen Toan Tran, and Angela Dawson. "The role of social capital in women’s sexual and reproductive health and rights in humanitarian settings: a systematic review of qualitative studies." Conflict and health 15.1 (2021): 1-12.

This conceptual framework should then guide which social capital items are introduced in the analysis. The questionnaire used in this study, which dates back to 2004 (World Bank) may be partly outdated. Current appraisals of the social capital literature all underline that precision in measurement is needed, because of the wide diversity of mechanism covered. The exact way in which the dimension is measured (the wording of the question) should be spelled out. The compound indicator "overall social capital" is also probably to be abandoned, for the same reason.

Finally, the analyses also show some gaps. First, why were age and sex, and other important covariates, not investigated and controlled for in relation to social capital? Second, the notes a, b and c .. in the multivariate analyses tables are not understandable. Also show the size of the effects for all the covariates, as a recurrent finding in the social capital health literature is that effect sizes are small.

6. PLOS authors have the option to publish the peer review history of their article (what does this mean?). If published, this will include your full peer review and any attached files.

**Do you want your identity to be public for this peer review?** For information about this choice, including consent withdrawal, please see our Privacy Policy.

Reviewer #1: No

Reviewer #2: **Yes: **Clémentine Rossier

---

## [Decision Letter · Decision Letter 1]

21 Jun 2023

PGPH-D-22-01611R1

The influence of social capital in the utilisation of sexual reproductive health services among the youth in Ghana. A community-based cross-sectional study.

Dear Dr. Ananga,

Thank you for submitting your manuscript to PLOS Global Public Health. After careful consideration, we feel that it has merit but does not fully meet PLOS Global Public Health’s publication criteria as it currently stands. Therefore, we invite you to submit a revised version of the manuscript that addresses the points raised during the review process.

Your manuscript has been reassessed by a new reviewer as unfortunately the reviewers from the previous round were not available. As you will see from the comments below, there remain some concerns which should be addressed before your manuscript is suitable for publication.

We look forward to receiving your revised manuscript.

Kind regards,

Joseph Donlan

Editorial Office

Journal Requirements:

Additional Editor Comments (if provided):

Reviewers' comments:

Reviewer's Responses to Questions

**Comments to the Author**

1. If the authors have adequately addressed your comments raised in a previous round of review and you feel that this manuscript is now acceptable for publication, you may indicate that here to bypass the “Comments to the Author” section, enter your conflict of interest statement in the “Confidential to Editor” section, and submit your "Accept" recommendation.

Reviewer #3: (No Response)

2. Does this manuscript meet PLOS Global Public Health’s publication criteria? Is the manuscript technically sound, and do the data support the conclusions? The manuscript must describe methodologically and ethically rigorous research with conclusions that are appropriately drawn based on the data presented.

Reviewer #3: Yes

3. Has the statistical analysis been performed appropriately and rigorously?

Reviewer #3: No

4. Have the authors made all data underlying the findings in their manuscript fully available (please refer to the Data Availability Statement at the start of the manuscript PDF file)?

Reviewer #3: Yes

5. Is the manuscript presented in an intelligible fashion and written in standard English?

Reviewer #3: Yes

6. Review Comments to the Author

Reviewer #3: 1. On the abstract, background section, stated that “Besides, while the factors associated with the use of reproductive health services among the youth are well documented in the literature, most studies have focused on aspects at the individual level.” It is good to examine the overlooked factors, but, the concept of social capital is not only at the societal level. Instead, it comprises at individual, community, institution levels.

2. On the abstract, conclusion section stated that “any intervention meant to address social capital needs in sexual and reproductive health should consider the efficacy of each social capital dimension and the intervention's environment.” What types of interventions are appropriate for youths to address their social capital needs and be specific!

3. You mentioned the reason why you select the study population. Please cite the reference that as evidence. “This age group was chosen because most youth within this group are currently sexually active. (Reference?)”

4. We cannot measure social capital directly because it is a latent variable. Why not you consider factor analysis that was appropriate to the variable?

5. Even though you have used a social capital questionnaire that was used in low income setting, the questionnaire might not be comprehensive enough to consider the context of Ghana. In addition, the tool was not validated in the study area and should be mentioned as a limitation of your study.

7. PLOS authors have the option to publish the peer review history of their article (what does this mean?). If published, this will include your full peer review and any attached files.

**Do you want your identity to be public for this peer review?** For information about this choice, including consent withdrawal, please see our Privacy Policy.

Reviewer #3: No

---

## [Decision Letter · Decision Letter 2]

24 Aug 2023

The influence of social capital in the utilisation of sexual reproductive health services among the youth in Ghana. A community-based cross-sectional study.

PGPH-D-22-01611R2

Dear Dr Ananga,

We are pleased to inform you that your manuscript 'The influence of social capital in the utilisation of sexual reproductive health services among the youth in Ghana. A community-based cross-sectional study.' has been provisionally accepted for publication in PLOS Global Public Health.

Best regards,

Julia Robinson

Executive Editor

Reviewer Comments (if any, and for reference):

Reviewer's Responses to Questions

**Comments to the Author**

1. If the authors have adequately addressed your comments raised in a previous round of review and you feel that this manuscript is now acceptable for publication, you may indicate that here to bypass the “Comments to the Author” section, enter your conflict of interest statement in the “Confidential to Editor” section, and submit your "Accept" recommendation.

Reviewer #3: All comments have been addressed

2. Does this manuscript meet PLOS Global Public Health’s publication criteria? Is the manuscript technically sound, and do the data support the conclusions? The manuscript must describe methodologically and ethically rigorous research with conclusions that are appropriately drawn based on the data presented.

Reviewer #3: Yes

3. Has the statistical analysis been performed appropriately and rigorously?

Reviewer #3: Yes

4. Have the authors made all data underlying the findings in their manuscript fully available (please refer to the Data Availability Statement at the start of the manuscript PDF file)?

Reviewer #3: Yes

5. Is the manuscript presented in an intelligible fashion and written in standard English?

Reviewer #3: Yes

6. Review Comments to the Author

Reviewer #3: (No Response)

7. PLOS authors have the option to publish the peer review history of their article (what does this mean?). If published, this will include your full peer review and any attached files.

**Do you want your identity to be public for this peer review?** For information about this choice, including consent withdrawal, please see our Privacy Policy.

Reviewer #3: No
